# Assessment of hepatitis B vaccination status and hepatitis B surface antibody titres among health care workers in selected public health hospitals in Kenya

**Irene Ann Mwangi** [1,2]*, **Jesca O. Wesongah** [3], **Victor Moses Musyoki** [2,4], **Gloria S. Omosa-Manyonyi** [2,4], **Bashir Farah** [2], **Laura Gwahalla Edalia** [5], **Margaret Mbuchi** [6]

1 Institute of Tropical Medicine and Infectious Diseases, Jomo Kenyatta University of Agriculture and Technology, Nairobi, Kenya, 2 KAVI–Institute of Clinical Research, University of Nairobi, Nairobi, Kenya, 3 Department of Medical Laboratory Sciences, College of Health Sciences, Jomo Kenyatta University of Agriculture and Technology, Nairobi, Kenya, 4 Department of Medical Microbiology and Immunology, School of Medicine, Faculty of Health Sciences, University of Nairobi, Nairobi, Kenya, 5 Department of Dental Sciences, Faculty of Health Sciences, University of Nairobi, Nairobi, Kenya, 6 Centre for Clinical Research, Kenya Medical Research Institute, Nairobi, Kenya

* wiann97@yahoo.com

## Abstract

Healthcare workers (HCWs) have a significant occupational risk of hepatitis B virus (HBV) infection. Vaccination remains the most effective measure recommended to avert the risk. However, there's limited information on hepatitis B vaccine uptake rates and the seroprotection status of HCWs, especially in sub-Saharan Africa. This study aimed to assess hepatitis B vaccination status and also seroprotection status of HCWs in three selected public hospitals in Kenya. This was a cross-sectional study carried out among HCWs at Kenyatta National Hospital (KNH), Naivasha and Mbagathi County hospitals. Data on participants' demographics and hepatitis B vaccination status was collected using an interviewer-guided questionnaire. Blood samples were collected and tested for hepatitis B surface antigen (HBsAg), hepatitis B surface antibodies (anti–HBs), and hepatitis B core antibodies (anti–HBc) using Enzyme Linked Immuno Sorbent Assay technique. Data were analyzed using Statistical Package for the Social Sciences (SPSS) and Graph pad prism. Of the 145 eligible HCWs, 120 (82.8%) were vaccinated, with 77 (53.1%) having received the recommended three doses. Three quarters (108/145) of the vaccinated HCWs were seroprotected (titres $\geq$10 mIU/ml) against HBV infection, while 16.6% were non–responders (titres <10 mIU/ml). Vaccination with more than two doses and HBV exposure were significantly associated with anti-HBs titre levels ($P$<0.05). HCWs who received less than 2 doses of the vaccine were 70% less likely to have high anti-HBs titre levels (aOR, 0.3; 95% CI, 0.1–0.8; $P$ = 0.013). Nearly all HCWs were vaccinated against hepatitis B virus. The majority of all HCWs were seroprotected against hepatitis B virus but a number of them had an insufficient immunity to the virus despite vaccination or prior exposure. There's need to sensitize HCWs and enforce mandatory full vaccination as per the recommended vaccination schedule.

**Data Availability Statement:** All relevant data are within the paper and its Supporting Information files.

**Funding:** The authors received no specific funding for this work.

**Competing interests:** The authors have declared that no competing interests exist.

## Introduction

Hepatitis, inflammation of the liver, remains a global public health concern and a significant cause of morbidity and mortality, especially in sub-Saharan Africa. Hepatitis B viral (HBV) infection is one of the leading causes of chronic hepatitis which progresses to cirrhosis and hepatocellular carcinoma [1]. The World Health Organization (WHO) estimates that 2 billion people worldwide have serological evidence of HBV infection, and that this illness causes 1 million deaths annually. Approximately 257 million people are chronically infected with majority residing in low and middle-income countries, particularly in Africa [1–4]. In Kenya, studies have reported Hepatitis B surface antigen prevalence rate ranging from 5% to 30% in the general population [5].

Chronically infected persons are the main reservoirs of HBV which is harbored in their blood and other body fluids like saliva, semen, vaginal secretions and tears with concentrations of HBV deoxyribonucleic acid being high in blood or serous exudates compared to mucosal secretions. Transmission is mainly through sexual intercourse, parenteral contact or mother to child during birth, depending on epidemiological patterns within geographical areas [3, 6].

Healthcare workers (HCW) are at a high risk of HBV infection through occupational exposure to blood and body fluids; the incidence is estimated to be 2–4 times the level in the general population [7]. Sub-Saharan Africa is considered to be a highly endemic area for HBV, with sero-prevalence among health workers in different countries ranging from 1.1% up to 17.8% [1].

As part of occupational safety measures, HCW are required to be vaccinated against HBV. The WHO has estimated that HBV vaccination coverage amongst health care workers is only 18%-39% in low and middle-income countries compared to 67%-79% in high-income countries [7]. It is estimated that 3 doses of Hepatitis B vaccine which consist of an initial vaccination, a repeat vaccination at one month, and another repeat vaccination at six month, induce protective antibody concentrations in >95% of healthy infants, children, and adolescents and in >90% of healthy adults [8]. Studies conducted in Sri Lanka, Rwanda and Northern Uganda show that proportion of vaccinated HCW with titres >10 mIU/ml ranged from 70–90% [9–11]. Anti–HBs titres levels ≥10 mIU/ml are considered to be protective against HBV infection. Individuals with antibody titres between 10 mIU/ml and 100 mIU/ml are categorized as hypo-responders, and hyper-responders if >100 mIU/ml. Some healthy people (5–15%) do not develop anti-HBs or respond poorly to the surface antigen component of HBsAg, and they are classified as non-responders (<10 mIU/ml). This population, especially among the HCWs due to exposure, increases the risk of infection and transmission. The Centers for Disease Control (CDC) recommends that for HCW, vaccination of the three dose series be followed by assessment of hepatitis B surface antibodies to determine immunological responses post vaccination, and if necessary, revaccination be done. Some of the associated factors documented for non-response to the Hepatitis B vaccine include older age, obesity, smoking and other chronic illnesses. Individuals who do not have protective anti-HBs levels ≥10 mIU/ml after revaccination (6 doses) ought to be tested for HBsAg and anti-HBc to determine their HBV infection status [9, 12–14].

Vaccination against HBV has been ongoing activity for several years in most of the countries including Kenya. However, there is limited information on vaccine uptake and seroprotection status against HBV, especially among HCWs in Kenya. This is partly attributed to the high cost of immunological testing and lack of a policy guideline that enforces mandatory and full vaccination of HCWs. This study therefore aimed to assess hepatitis B vaccination status and seroprotection status of Health Care Workers (HCW) at three selected public hospitals in Kenya as well as determine factors associated with anti-HBs immune responses.

## Methods

### Study design, setting and population

This was a cross sectional study with sub group analysis conducted through consecutive sampling of heath care workers (medical doctors, nurses and laboratory technologists) at Kenyatta National Hospital (KNH), a national and teaching referral hospital based in Nairobi, Naivasha sub-County hospital based in Nakuru County and Mbagathi County Hospital based in Nairobi. One hundred and forty eight heath care workers (HCW) who were in direct contact with patients or patients' samples were recruited within a period of three months.

### Data and samples collection

Medical and socio-demographic data was collected using an interviewer-guided questionnaire (S1 Appendix) after obtaining written informed consent. Health care workers who had received one or more hepatitis B vaccine doses were considered to be vaccinated; those who received the three recommended doses were considered to be fully vaccinated. Using Beckton Dickson serum separating tube (BD SST), 5mls of blood sample was collected from health care workers by a trained nurse. Samples from KNH and Mbagathi county hospitals were transported in a cool box to Kenya AIDs Vaccine Initiative–Institute of Clinical Research (KAVI–ICR), University of Nairobi for serum separation and storage while those from Naivasha sub-county hospital were centrifuged on site and serum transported to KAVI-ICR in dry ice. Centrifugation of all whole blood samples was done for 10 minutes at 3000 revolutions per minute, serum aliquoted into 2mls sterile Sarstedt screw cap microtubes and stored at -80˚C awaiting analysis.

### Laboratory serological assays

All assays were performed at KAVI–ICR using validated commercially available Enzyme Linked Immunosorbent Assay (ELISA) kits as per the manufacturer's instructions. Screening for chronic Hepatitis B virus (HBV) infection on the serum samples marked by presence of hepatitis B surface antigen (HBsAg) was performed using BioELISA HBsAg kit version 3.0 (Werfen, Barcelona, Spain). Three (3) participants' samples that tested positive for HBsAg were excluded from the study. Samples that tested negative for HBsAg (145) were further analyzed using BioELISA anti-HBs ELISA kit to assess for presence and concentration of antibodies against hepatitis B surface antigen (anti-HBs) as a response to vaccines or natural infection; seroprotection was defined by anti-HBs titres $\geq$10 mIU/ml. The samples were also tested for anti-HBc, a marker for natural infection using BioELISA anti-HBc kit, as per the manufacturer's instructions. Hepatitis B virus (HBV) exposure was defined by positive serological evidence of natural infection determined by a positive anti-HBc test. Naïve HCWs who were unvaccinated and unexposed to HBV were excluded from immune response distribution as they had not been immunologically challenged. Study participants were then classified into three groups based on anti-HBs immune response titres: non-responsive ($<$10 mIU/ml), hypo-responsive (10–100 mIU/ml) and hyper-responsive ($>$100 mIU/ml) according to Center for Disease control and Prevention (CDC) guidelines, where protection is defined as anti-HBs titres greater than 10 mIU/ml.

### Statistical analysis

Data analysis was done using IBM SPSS statistics version 21 and Graph pad prism version 9.0. In univariate analysis, frequency distribution and proportions was used for categorical variables such as gender, healthcare facility, vaccination and anti-HBS serological status. Pearson

chi square was used in bivariate analysis to determine any association between anti-HBs status with vaccination and HBV exposure status. A p-value of <0.05 was considered statistically significant. Multivariate analysis was used to determine and estimate any relationship and strength between predictor variables (age, sex, number of vaccine doses received, anti–HBc status) and the outcome variable (anti–HBs status).

### Ethical considerations

This study was approved by KNH-UoN Ethics and Research Committee (P330/04/2016). Permission to conduct the study was granted by KNH, Naivasha sub-county and Mbagathi hospitals. Participants were sensitized on the study and informed of potential risks and benefits after which a written informed consent was obtained. No personal identifiers were used during data, sample collection and analysis. All participants were given a unique identifier and key records identifying a participant were kept confidential by the principal investigator.

## Results

### Demographic characteristics and hepatitis B vaccination status of study participants

A total of 148 HCWs were recruited from KNH, Naivasha sub-county and Mbagathi hospitals. Out of these, three tested positive for HBsAg and were excluded from the study. Of the remaining 145 participants, 85 (58.6%) participants were female. Among the participants, 22 (15.2%) were medical doctors, 80 (55.2%) were laboratory technologists and 43 (29.7%) were nurses. Majority of the participants (104, 71.7%) were from Kenyatta National Hospital. One hundred and twenty (82.8%) participants had received at least one dose of the hepatitis B vaccine, 77 (53.1%) were fully vaccinated with the recommended three doses and 4 (3.3%) had received a booster dose. One hundred and thirteen (94.2%) of the vaccinated HCW were not tested for anti–HBs response post vaccination (Table 1).

### Distribution of anti-HBs responses among HCWs at KNH, Mbagathi county and Naivasha sub-county hospitals

One hundred and one (84.2%) of the vaccinated HCWs and 9 (36.0%) of the unvaccinated had hepatitis B surface antibodies. Twenty-eight (90.3%) of 31 participants with serological evidence of HBV exposure (anti–HBc positive) had hepatitis B surface antibodies. Nineteen (15.8%) of the vaccinated and 3 (9.7%) of the HBV exposed had no detectable hepatitis B surface antibodies. Presence of anti–HBs titres was significantly higher among the vaccinated (*P*<0.001) and HBV exposed (*P* = 0.034) compared to the unvaccinated unexposed (Table 2).

Ninety seven (80.8%) of the vaccinated HCWs did not have serological evidence of HBV exposure. Among the 31 HBV exposed, 23 (74.2%) were reported to have received a hepatitis B vaccine.

Immune response for 128 HCWs vaccinated or HBV exposed was assessed and classified based on anti–HBs titres responder types. Eighteen (78.3%) vaccinated HCWs with serological evidence of HBV exposure and 69 (71.1%) vaccinated with no evidence of exposure were hyper-responders (>100 mIU/ml). Six (75%) of the HBV-exposed but unvaccinated HCW were also hyper-responders. Twenty (16.6%) of the vaccinated participants were non-responders with titres <10 mIU/ml. Nineteen of these had no detectable anti-HBs, while one had titres less than 10 mIU/ml. Twelve of the twenty non-responders had only received one dose of the vaccine, eight had received two doses, and three had been exposed to HBV (Table 3).

**Table 1. Demographic characteristics and hepatitis B vaccination status of participating HCW at KNH, Mbagathi County and Naivasha sub-county hospitals (N = 145).**

| Characteristics | Description | n (%) |
|---|---|---|
| **Age (years)** | <25 | 31 (21.4%) |
| | 26–35 | 39 (26.9%) |
| | 36–45 | 32 (22.1%) |
| | 46–55 | 28 (19.3%) |
| | >56 | 15 (10.3%) |
| **Sex** | Female | 85 (58.6%) |
| | Male | 60 (41.4%) |
| **Cadre** | Medical doctor | 22 (15.2%) |
| | Laboratory Technologist | 80 (55.2%) |
| | Nurse | 43 (29.7%) |
| **Study site** | Kenyatta National Hospital (KNH) | 104 (71.7%) |
| | Naivasha Sub-County Hospital | 29 (20.0%) |
| | Mbagathi County Hospital | 12 (8.3%) |
| **Vaccination status** | Vaccinated | 120 (82.8%) |
| | Unvaccinated | 25 (17.2%) |
| **Vaccine doses received** | 1 | 9 (6.2%) |
| | 2 | 29 (20.0%) |
| | 3 | 77 (53.1%) |
| | 4 | 5 (3.4%) |
| | 0 | 25 (17.2%) |
| **Anti–HBs Test** | Tested | 5 (4.2%) |
| **Post Vaccination** (n = 120, vaccinated) | Not tested | 113 (94.2%) |
| | Missing response | 2 (1.7%) |

## Factors associated with anti-HBs titre immune responses among HCWs at KNH, Mbagathi county and Naivasha sub-county hospitals

Multinomial logistic regression analysis showed that HCWs who received less than 2 doses of the vaccine were 70% less likely to have anti-HBs (aOR, 0.3; 95% CI, 0.1–0.8; $P$ = 0.013). Females were 2.8 times more likely to have anti-HBs compared to males (aOR, 2.8; 95% CI, 1.0–7.9; $P$ = 0.056). Age and HBV exposure were not determinants of immune response (Fig 1).

## Discussion

The aim of this study was to assess hepatitis B vaccination status and seroprotection status of Health Care Workers (HCW) as well as determine factors associated with anti-HBs immune responses.

**Table 2. Anti–HBs response based on vaccination and HBV exposure status among HCW at KNH, Mbagathi County and Naivasha sub-county hospitals.**

| Characteristic | Description | n (%) | Anti–Hepatitis B status (N = 145) | | P Value |
|---|---|---|---|---|---|
| | | | Anti–HBs Positive n (%) | Anti–HBs Negative n (%) | |
| **Vaccination status** | Vaccinated | 120 (100%) | 101 (84.2%) | 19 (15.8%) | P<0.001 |
| | Unvaccinated | 25 (100%) | 9 (36.0%) | 16 (64.0%) | |
| **HBV Exposure status** | Anti–HBc Positive | 31 (100%) | 28 (90.3%) | 3 (9.7%) | P = 0.034 |
| | Anti–HBc Negative | 114 (100%) | 82 (71.9%) | 32 (28.1%) | |

**Table 3. Classification of immune responses based on anti-HBs titres, vaccination and HBV exposure status among HCW at KNH, Mbagathi County and Naivasha sub-county hospitals.**

| Responder type (Anti–HBs titres) | Vaccinated | | Unvaccinated |
|---|---|---|---|
| | HBV Exposed (n = 23) | HBV Unexposed (n = 97) | HBV Exposed (n = 8) |
| <10 mIU/ml | 3 (13.0%) | 17 (17.5%) | 0 (0.0%) |
| 10–100 mIU/ml | 2 (8.7%) | 11 (11.3%) | 2 (25.0%) |
| >100 mIU/ml | 18 (78.3%) | 69 (71.1%) | 6 (75.0%) |

Nearly all the HCWs had received at least one dose of the vaccine with half of the study participants having been fully vaccinated. The high percentage of vaccine uptake in this study concurs with the findings of a similar study conducted among HCWs in Makueni County, Kenya, where 80% of the HCWs had received at least one dose of the vaccine and 48% had completed the three recommended doses [15]. These findings show an improvement from an earlier report on HCWs in Kenya where vaccine uptake was 13% [16]. However, the finding is lower compared to similar studies conducted among HCW in Sri Lanka, Rwanda and Northern Uganda where completion of the three dose hepatitis B regimen ranged between 65–96% [9–11]. The improvement in vaccine uptake in Kenya could be attributed to high sensitization on need of vaccination amongst HCWs as well as provision of vaccines by employers.

Hepatitis B vaccination and assessment of antibody levels is the most effective way of reducing incidences of HBV infection and limiting transmission. Overall, three quarters of all HCWs in this study had detectable anti–HBs titres that conferred protection against HBV infection. Majority of vaccinated HCWs had protective titres ≥10 mIU/ml with almost three quarters being hyper–responders (>100 mIU/ml). This compares to similar studies among health care workers in Sri Lanka, Rwanda and Northern Uganda where proportion of vaccinated HCW with titres ≥10 mIU/ml ranged from 70–90% [9–11]. Contrary to our finding, a

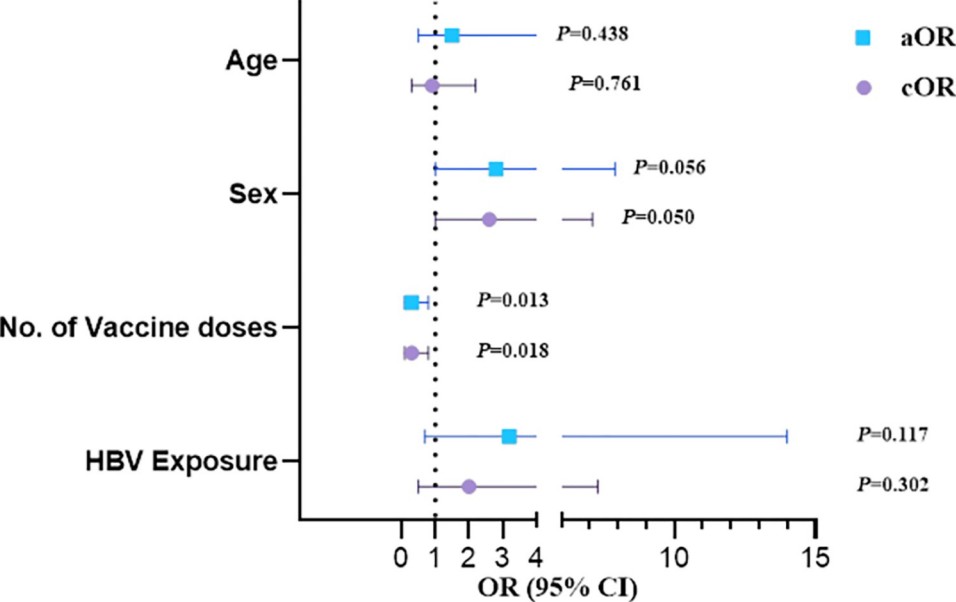

**Fig 1. Odds ratio for covariates that impact anti-HBs production among HCW.** ORs and 95% CI with significant associations indicated at P<0.05 Abbreviations: OR, odds ratio; cOR, crude odds ratio; aOR, adjusted odds ratio; CI, confidence interval.

study by Sabina Sernia et.al in an large Italian university revealed that only 49.6% of students in health related professions had protective antibody titres [17]. The high proportion of HCWs with protective anti–HBs titres in this study is an indicator of effective immune responses to hepatitis B vaccine.

A higher percentage of immune response was observed in HCWs who had received more than two doses of Hepatitis B vaccine, a finding that concurs with a study in Rwanda that reported similar responses among HCWs who had received two and three doses of the vaccine [9]. Findings from the Italian study by Sabina Sernia et.al concur with the findings of this study as it revealed that immune responses were higher in health profession students who had received 3–4 doses of the vaccine [17].

Literature on vaccine and immunology has shown that vaccination, the act of introducing the vaccine, does not always lead to a positive immune response (immunization). In this study, almost a fifth of the HCWs did not have sufficient protective titres post vaccination a finding that agreed with the reported global level of poor immune response to HBV vaccination of 5–15% [9, 12, 13]. In contrast, our finding was lower compared to a study among blood donors in Italy that reported 28% non-response [18]. The high percentage of non-response was of concern as the study revealed that almost all of the vaccinated HCWs did not test for immune responses post vaccination and therefore non-responders were not aware of their immune status hence high susceptibility to HBV infection.

Sex specific differences influence immune responses to vaccines. Our study revealed that females were more likely to respond immunologically to the vaccine compared to males and this concurs with findings from similar studies conducted in different parts of the world among health care workers that reported females as high responders to vaccine compared to males. A study by Trevisan and colleagues on sex disparity in response to Hepatitis B vaccine reported that females showed 20% higher vaccination-related antibody titres than males [19]. Similarly, two independent studies on HBV vaccination and immunization from India and Rwanda among health care workers reported a high non-response rate to vaccination among males (13.7–16%) compared to females (5.3–7.9%) [9, 10]. Contrary to these findings, Basireddy et.al study in India showed no association between sex and immune seroconversion post vaccination [20]. The high immune response among females reported in this and other similar studies could be attributed to smoking, alcohol and certain genetic factors. However, these factors especially the socio-behavioral factors which are also associated with females were not assessed in these studies, including the present study [9, 10, 19].

Exposure to HBV triggers the immune system to produce antibodies. In this study, majority of HCWs exposed to HBV were seroprotected although three quarters were also vaccinated. These findings are consistent with similar studies on responses to hepatitis B vaccine in isolated anti-HBc positive adults carried out in China, Korea and Uganda that reported immune response ranging between 86%–92% [11, 21]. Presence of anti–HBs and anti–HBc is a serological pattern mostly seen in individuals who have naturally acquired immunity to HBV infection and this is most common in high risk groups. Among the 31 HBV exposed HCWs, a tenth did not have hepatitis B surface antibodies and this could be an indication of a distant resolved HBV infection [4]. We could not establish if the HBV exposure among the vaccinated was breakthrough or they had been exposed before vaccination.

A study conducted in Kenya among HCWs utilizing pre–vaccination testing of anti–HBs to determine vaccination requirements revealed that assessment of serological status prior to vaccination is cost effective as it identified individuals who were already exposed and had naturally acquired immunity hence did not require vaccination [16].

The main limitation in this study was that vaccination status was self-reported since participants did not have vaccination cards. However, study participants were requested to provide the year they received their last dose of vaccine.

## Conclusion

Nearly all health care workers were vaccinated against Hepatitis B Virus. The majority of all health care workers were seroprotected against hepatitis B virus but a number of them had an insufficient immune response to the virus despite vaccination or prior exposure. Additionally, majorities of HCWs were not tested for anti–HBs status post vaccination and were not aware of their immune status against HBV infection.

These findings highlight the need for the government to impose measures to sensitize HCWs and enforce mandatory full vaccination as per the recommended vaccination schedule which should be followed by confirmation of serological status post vaccination to ensure safety and reduce transmission rates. We further recommend that the government vaccination programs consider documentation of HBV vaccinations received for future reference.

## Supporting information

**S1 Appendix. Interviewer-guided questionnaire.**
(PDF)

**S1 Data.**
(XLSX)

## Acknowledgments

We wish to acknowledge the training and support from 'UANDISHI-Building Capacity for Writing Scientific Manuscripts' program at the Faculty of Health Sciences, University of Nairobi. The contents of this manuscript are the responsibility of the authors and do not necessarily reflect the views of USAID or the U.S. Government.

## Author Contributions

**Conceptualization:** Irene Ann Mwangi, Jesca O. Wesongah, Bashir Farah, Margaret Mbuchi.

**Formal analysis:** Irene Ann Mwangi, Victor Moses Musyoki.

**Investigation:** Irene Ann Mwangi, Jesca O. Wesongah, Margaret Mbuchi.

**Methodology:** Irene Ann Mwangi, Bashir Farah.

**Resources:** Irene Ann Mwangi.

**Supervision:** Jesca O. Wesongah, Margaret Mbuchi.

**Validation:** Irene Ann Mwangi.

**Writing – original draft:** Irene Ann Mwangi.

**Writing – review & editing:** Irene Ann Mwangi, Jesca O. Wesongah, Victor Moses Musyoki, Gloria S. Omosa-Manyonyi, Bashir Farah, Laura Gwahalla Edalia, Margaret Mbuchi.

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
