## [Decision Letter · Decision Letter 0]

2 Dec 2022

PGPH-D-22-01550

Assessment of Hepatitis B vaccination status and Hepatitis B surface antibody titres among health care workers in selected public health hospitals in Kenya

Dear Dr. Mwangi,

Thank you for submitting your manuscript to PLOS Global Public Health. After careful consideration, we feel that it has merit but does not fully meet PLOS Global Public Health’s publication criteria as it currently stands. Therefore, we invite you to submit a revised version of the manuscript that addresses the points raised during the review process.

The manuscript has been evaluated by three reviewers, and their comments are available below.

The reviewers have raised a number of concerns that need attention. They request additional information on the difference between immunization status and vaccination status, on methodological aspects of the study (blood sample preparation), and they feel an important finding of your study was not addressed in the discussion. 

Could you please revise the manuscript to carefully address the concerns raised?

We look forward to receiving your revised manuscript.

Kind regards,

Alex Schaefer, PhD

Staff Editor

Journal Requirements:

Reviewers' comments:

Reviewer's Responses to Questions

**Comments to the Author**

1. Does this manuscript meet PLOS Global Public Health’s publication criteria? Is the manuscript technically sound, and do the data support the conclusions? The manuscript must describe methodologically and ethically rigorous research with conclusions that are appropriately drawn based on the data presented.

Reviewer #1: Yes

Reviewer #2: Yes

Reviewer #3: Yes

2. Has the statistical analysis been performed appropriately and rigorously?

Reviewer #1: No

Reviewer #2: Yes

Reviewer #3: Yes

3. Have the authors made all data underlying the findings in their manuscript fully available (please refer to the Data Availability Statement at the start of the manuscript PDF file)?

Reviewer #1: Yes

Reviewer #2: Yes

Reviewer #3: Yes

4. Is the manuscript presented in an intelligible fashion and written in standard English?

Reviewer #1: Yes

Reviewer #2: Yes

Reviewer #3: Yes

5. Review Comments to the Author

Reviewer #1: authors should address all the points to consider acceptance for publication. i recommend to focus the immunization status than vaccination status. Theoretical, there is no immunization without the exposure for an antigen

Reviewer #2: Diligent study and HCW has to give great care while performing their duties because it is hazardous condition that should be treated within time and HCW are at great risk as they are exposing by the time to the patients

Government has to take action for taking care of vaccination of HCW

Reviewer #3: I am grateful for the opportunity to review the manuscript entitled “Assessment of Hepatitis B vaccination status and Hepatitis B surface antibody titres among health care workers in selected public health hospitals in Kenya.” The authors have assessed the hepatitis vaccination and seroprotection status of health care workers in the Kenyan population. This study is important because healthcare workers are more prone to hepatitis B infection than the general populace and there is a need to continue to sensitize health care workers and encourage full vaccination according to the recommended vaccination schedule. However, the following minor comments need to be addressed:

Methods

P.5, lines 103-104, kindly give the detail of how serum was separated from the whole blood samples collected from study participants. What centrifugation speed was used, for how long did you spin and what kind of bottles were the serum samples aliquoted into before storage at -80 ºC?

Results

P.10, lines 185-186, “Females were 2.8 times more likely to have anti-HBs compared to males.” This is an important finding which was not included in the discussion section. The authors need to explain why females were more likely to have anti-HBs compared to males in the study.

Discussion

P.13, line 226, please correct “The high percentages of non-response was concern as the study revealed” to “The high percentages of non-response was of concern as the study revealed”

6. PLOS authors have the option to publish the peer review history of their article (what does this mean?). If published, this will include your full peer review and any attached files.

**Do you want your identity to be public for this peer review?** For information about this choice, including consent withdrawal, please see our Privacy Policy.

Reviewer #1: **Yes: **Aschalew Afework Bitew

Reviewer #2: **Yes: **Dr Raheela Afzal

Reviewer #3: **Yes: **Subulade A. Ademola

---

## [Decision Letter · Decision Letter 1]

1 Mar 2023

Assessment of Hepatitis B vaccination status and Hepatitis B surface antibody titres among health care workers in selected public health hospitals in Kenya

PGPH-D-22-01550R1

Dear Ms Mwangi,

We are pleased to inform you that your manuscript 'Assessment of Hepatitis B vaccination status and Hepatitis B surface antibody titres among health care workers in selected public health hospitals in Kenya' has been provisionally accepted for publication in PLOS Global Public Health.

Best regards,

Julia Robinson

Executive Editor

Reviewer Comments (if any, and for reference):

Reviewer's Responses to Questions

**Comments to the Author**

1. If the authors have adequately addressed your comments raised in a previous round of review and you feel that this manuscript is now acceptable for publication, you may indicate that here to bypass the “Comments to the Author” section, enter your conflict of interest statement in the “Confidential to Editor” section, and submit your "Accept" recommendation.

Reviewer #1: (No Response)

Reviewer #2: All comments have been addressed

2. Does this manuscript meet PLOS Global Public Health’s publication criteria? Is the manuscript technically sound, and do the data support the conclusions? The manuscript must describe methodologically and ethically rigorous research with conclusions that are appropriately drawn based on the data presented.

Reviewer #1: Partly

Reviewer #2: Yes

3. Has the statistical analysis been performed appropriately and rigorously?

Reviewer #1: Yes

Reviewer #2: Yes

4. Have the authors made all data underlying the findings in their manuscript fully available (please refer to the Data Availability Statement at the start of the manuscript PDF file)?

Reviewer #1: Yes

Reviewer #2: Yes

5. Is the manuscript presented in an intelligible fashion and written in standard English?

Reviewer #1: Yes

Reviewer #2: Yes

6. Review Comments to the Author

Reviewer #1: my worry is that there is a big difference between 'vaccination and immunization'. However, authors used these two different words interchangeably. This confuses the readers and also it will be difficult to understand the document as these two words are critical for this manuscript

Reviewer #2: (No Response)

7. PLOS authors have the option to publish the peer review history of their article (what does this mean?). If published, this will include your full peer review and any attached files.

**Do you want your identity to be public for this peer review?** For information about this choice, including consent withdrawal, please see our Privacy Policy.

Reviewer #1: **Yes: **Aschalew Afework Bitew

Reviewer #2: **Yes: **Dr. Raheela Afzal
